# Research on Greenhouse Gas Emissions and Economic Assessment of Biomass Gasification Power Generation Technology in China Based on LCA Method

**Yuan Wang * and Youzhen Yang**

School of Finance, Shanxi University of Finance and Economics, Taiyuan 030006, China
* Correspondence: 20201024@sxufe.edu.cn

**Abstract:** China is rich in biomass resources, taking straw as an example, the amount of straw in China is 735 million tons in 2021. However, at the level of resource use, biomass resources have the practical difficulties of being widely distributed and difficult to achieve large-scale application. By collecting large amounts of biomass and generating electricity using gasification technology, we can effectively increase the resource utilization of biomass and also improve China's energy security. By using a life cycle assessment (LCA) approach, this paper conducted a life cycle assessment with local biomass gasification power generation data in China and found that the LCA greenhouse gas emissions of biomass gasification power generation technology is 8.68 t $CO_2$ e/$10^4$ kWh and the LCA cost is 674 USD/$10^4$ kWh. Biomass gasification power generation technology has a 14.7% reduction in whole life carbon emissions compared to coal power generation technology. This paper finds that gas-fired power generation processes result in the largest carbon emissions. In terms of economics, this paper finds that natural gas brings the most additional costs as an external heat source.

**Keywords:** life cycle assessment (LCA); biomass gasification for power generation; carbon emissions; economics

## 1. Introduction

Since the Paris Agreement, countries around the world have been diversifying and decarbonizing their energy supply in response to the extreme climate problems caused by greenhouse gases [1]. In particular, the proportion of renewable energy sources is increasing as a percentage of global primary energy. Global renewable energy share of the electricity mix reached 12.8% in 2021 [2]. Among the renewable energy sources, biomass resources have become the most popular green energy source in the 21st century because of their abundant resources, wide distribution, and environmental friendliness [3].

Biomass energy is utilized in the form of oxidative combustion, thermochemical conversion, compression reforming, and biomass conversion [4]. Biomass gasification technology for power generation is the application of biomass in the field of biomass conversion. The solid biomass is heated in a gasifier and passed through air and water vapor to produce combustible gas, which is then passed through a gas turbine to produce electricity.

China's biomass resources mainly originate from the agricultural sector, with straw, for example, amounting to 735 million tons in 2021 [5] with a comprehensive utilization rate of 88%. The biomass resources available for exploitation in China are about 300 million tons of standard coal per year [6]. However, at the level of resource use, biomass resources are widely distributed and difficult to achieve large-scale applications. Biomass gasification power generation technology, which can transport China's highly dispersed biomass resources in the form of gas and then generate electricit power generation technology, can effectively promote the centralized and large-scale application of biomass resources [7]. Biomass gasification power generation technology not only reduces the proportion of wasted biomass resources, but also generates certain low carbon benefits. It is a green

technology that has been highly valued and encouraged by the Chinese government as it can effectively contribute to the realization of China's strategy of "achieving peak carbon by 2030 and carbon neutrality by 2060" (double carbon).

The current biomass gasification technology still suffers from problems such as higher costs from the application level, in addition, from a whole life cycle perspective, the whole chain process from biomass collection to biomass power generation can also lead to greenhouse gas emissions [8].In order to further promote the large-scale application of this technology in China, it is of great academic significance and application value to select the economic level as well as the greenhouse gas emission level for technology assessment from a whole life cycle perspective.

In the evaluation of biomass gasification technology for power generation, scholars have achieved certain research results. Wang Wei analyzed the energy consumption and environmental impact of 1 MW and 5.5 MW biomass gasification power systems using life cycle analysis, but this study did not adequately measure carbon emissions and did not consider the replacement cost of biomass power [9]. Shafie compared the technology of rice straw power generation with coal power generation and found that the former reduced GHG emissions by 1.79 kg $CO_2$/kwh compared to the latter [10]. Dias used LCA assessment methods and found that direct combustion of Canadian short rotary willow reduced GHG emissions by 85% compared to fossil fuels [11]. J.R. Nunes used the LCA assessment method to evaluate and analyze biomass power generation technologies for grate and fluidized bed furnaces in the Portuguese region and found that both technologies have a high sensitivity on water content and plant lifetime [12]. Song evaluated biomass gasification power generation technology for municipal waste in Macao and found that its GHG emissions were about 0.95 kg $CO_2$/kwh [13]. Liu et al. used input-output methods for basic data collection to assess the energy consumption and emissions of biomass power systems in China, and found that the carbon emissions during power generation were the largest [14]. However, this assessment method is to apply the price quantity to invert the obtained energy consumption quantity, and there is a certain error with the actual biomass power generation whole life cycle process.

Carpentieri used the LCA method to evaluate the combined biomass gasification and $CO_2$ removal technologies and found that the environmental impact due to plant construction is almost negligible in this process step. However, the list of LCA data in this paper is not sufficient, e.g., it does not include data for the biomass growing phase [15]. Tonini used the LCA approach to explore future energy use scenarios in Denmark using biomass as the energy base. Biofuels are mainly used in heavy land transport, ships, defense and aviation, and to optimize the Danish electricity mix through biomass power generation. According to the estimation, the current Danish biomass resources are not sufficient to support the energy use scenarios presented in the paper, so additional crops are needed [16]. Gerber modeled a multi-objective system for synthetic natural gas and electricity from biomass and performed the corresponding thermoeconomic analysis [17]. Sebastián applied the LCA method to study the GHG emissions of biomass combustion power technology and found that the biomass generation efficiency is the most important factor affecting the GHG emissions of LCA for this technology. However, this work did not include the process of biomass recovery in the LCA boundary setting [18]. Wang used a hybrid model of input-output approach and LCA to assess the uncertainty of future development of biomass power technology in China was studied and found that supply chain matching and power generation technology are the causes of uncertainty [19].

Related studies have shown that the resource utilization of agricultural waste can effectively promote the process of carbon peaking and carbon neutrality in China. The whole life assessment studies with local biomass power generation data in China are still very few, and the only studies focus on the analysis of municipal waste. Since rural energy data are difficult to collect, the LCA measurements are estimated by the input-output method, and therefore the evaluation conclusions obtained need further justification.

This paper collects the data of China's indigenous biomass gasification and power generation technology covering the whole process of biomass cultivation, biomass collection, biomass transportation and biomass gasification and power generation, and performs LCA calculations to be able to objectively assess the contribution potential of this technology to the reduction of GHG emissions, as well as the additional costs that may arise. By decomposing and measuring the contribution of greenhouse gas emissions and economics of biomass gasification power generation technology, this paper is able to dissect the key areas of biomass gasification power generation technology with larger emissions and make process recommendations. Through LCA measurements, this paper is able to derive formulas for estimating the cost of biomass gasification power generation technologies in China, which can provide an important academic basis for exploring the modeling process and evaluation process of non-fossil energy-based power generation technologies and their economic effects. Finally, the research in this paper is of high research value as it can contribute to the realization of the process of resource utilization in the rural areas of China as well as the achievement of carbon peaking and carbon neutrality goals in rural areas of China.

## 2. Methodology

The Life Cycle Assessment (LCA) methodology used in this paper is a methodological approach to the material and energy inputs and environmental loads involved in the production of a unit of product "from cradle to grave". LCA consists specifically of the setting of assessment boundaries, inventory analysis, and assessment analysis [20]. The functional unit of the biomass gasification power generation process studied in this paper is $10^4$ kwh of electricity, and the LCA accounting boundary for biomass gasification power generation is shown in Figure 1.

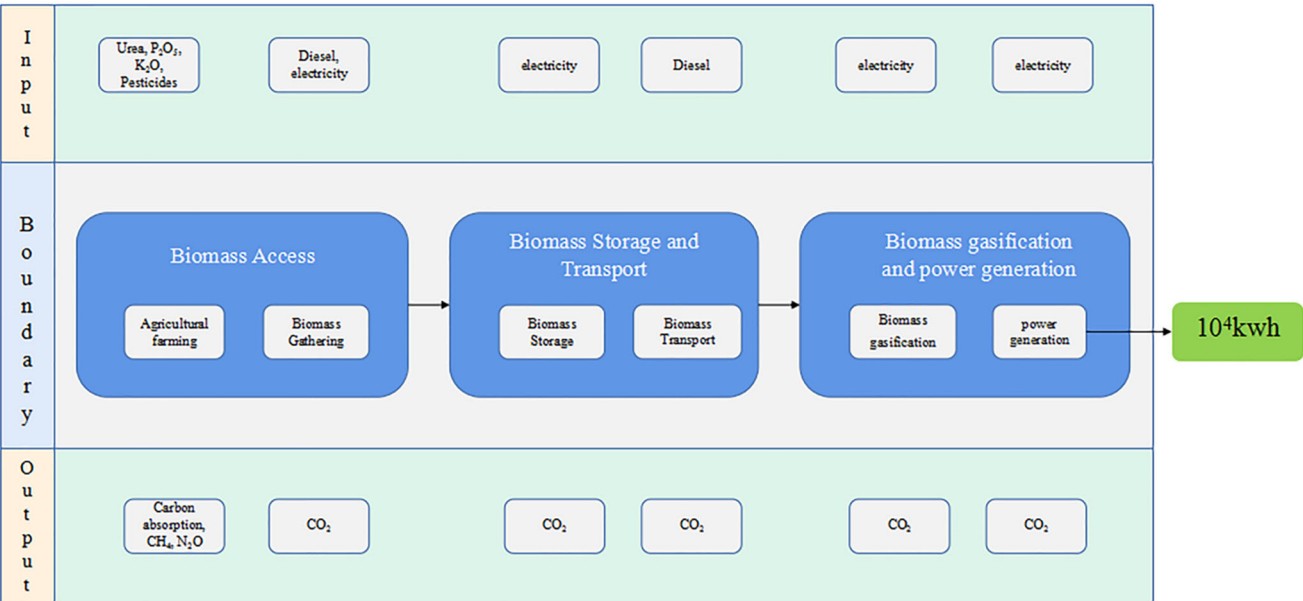

**Figure 1.** LCA assessment boundary for biomass gasification power generation technology.

As shown in Figure 1, in order to obtain $10^4$ kWh of biomass electricity, three stages are required: biomass access, biomass storage and transport, and biomass gasification and power generation. In the whole life cycle process, the material and energy inputs are reflected in the "input", and the greenhouse gas emissions such as $CO_2$ are reflected in the "output".

### 3. Data Collection and Inventory Analysis

In accordance with the full life cycle boundary of biomass gasification and power generation technology in this paper, this section provides a full life cycle inventory accounting for the three stages of biomass access, biomass transportation and storage, and biomass gasification and power generation. In the data inventory, the specific aspects of each process, energy consumption species, energy consumption, emission gas species, greenhouse gas emissions, and costs are included.

(1)  Biomass Access

The biomass access phase consists of two processes: agricultural farming and biomass collection (Table 1).

**Table 1.** Biomass access phase life cycle inventory data.

| | Sessions | Energy Consumption Species | Energy Input/MJ | Emission Content | Greenhouse Gas Emissions/kg $CO_2$ e | Cost/USD |
|---|---|---|---|---|---|---|
| | Mineral Mining | Diesel | 62 | $CO_2$ | 5.3 | 2.4 |
| | Mineral transportation | Diesel | 101 | $CO_2$ | 8.6 | 3.9 |
| | Urea production | Electricity | 3021 | $CO_2$ | 487.5 | 75.5 |
| | $P_2O_5$ production | Electricity | 158 | $CO_2$ | 25.6 | 4 |
| | $K_2O$ production | Electricity | 34 | $CO_2$ | 5.5 | 0.8 |
| Agricultural farming [21] | Pesticide production | Electricity | 73.52 | $CO_2$ | 11.9 | 1.8 |
| | Agricultural transportation | Diesel | 49.43 | $CO_2$ | 4.2 | 0.6 |
| | Crop cultivation | Diesel | 510 | $CH_4$ | 544.4 | 18.6 |
| | | | | $N_2O$ | 48.96 | |
| | | | | $CO_2$ absorption | −238.4 | |
| Biomass collection [21] | Mechanical Collection | Electricity | 74 | $CO_2$ | 12 | 1.9 |
| | Vehicle transportation | Diesel | 1120 | $CO_2$ | 93.9 | 40.7 |

Note: The cost price calculation unit price is calculated according to "Diesel price: 1.2 USD/L, 1 kwh electricity price: 0.09 USD/L, 1 m$^3$ natural gas price: 0.55 USD/m$^3$". The cost expenses such as plant construction cost and labor input are not included.

In order to generate $10^4$ kWh of biomass electricity, about 16 t of rice husk is needed. [14]. It is assumed that the proportion of rice husk in the rice crop is 5%. In the process of rice cultivation, 1.9 L of diesel fuel is required for mineral production, and according to IPCC, the carbon emission factor of 1 L of diesel fuel is 2.77 $CO_2$ e/L [22], Therefore, the carbon emissions emitted are 5.3 kg $CO_2$ e; Transporting the minerals to the urea and fertilizer processing plant requires the consumption of 3.1 L of diesel fuel, resulting in emissions of 8.6 L of carbon emissions, resulting in 8.6 kg of $CO_2$ e; In the urea and fertilizer processing plant, 839.1 kWh of electricity is required to produce urea using the aqueous solution full cycle method, and the annual carbon emission factor of 0.581 t $CO_2$/MWh [23] in China in 2021 is measured, and the carbon emission from urea production is 487.5 kg $CO_2$ e; In the $P_2O_5$ production phase, 44 kWh of electricity input is required, resulting in carbon emissions of 25.6 kg $CO_2$ e; In the $K_2O$ production phase 9.4 kWh of electricity input is required, resulting in carbon emissions of 5.5 kg $CO_2$ e; In the pesticide production phase, 20.4 kWh of electricity input is required, resulting in a carbon emission of 11.9 kg $CO_2$ e; Transporting produced agricultural materials such as pesticides and fertilizers to the rice growing area by truck, a process that consumes 1.5 L of diesel and emits 4.2 kg of $CO_2$ e; The use of agricultural machinery for rice cultivation consumes about 15.5 L of diesel fuel,

resulting in 42.9 kg $CO_2$ e of indirect carbon emissions, while this process results in the emission of 544.4 kg $CO_2$ e of $CH_4$ and 48.96 kg $CO_2$ e of $N_2O$ due to the use of chemical fertilizers and the sequestration of 238.4 kg $CO_2$ e of carbon due to plant photosynthesis, resulting in total net emissions in this phase of 398 kg $CO_2$ e. In order to collect 16 t of biomass, about 20.6 kwh of electricity is used, resulting in 12 kg $CO_2$ e of carbon emissions, and about 33.9 L of diesel fuel is used, resulting in 93.9 kg $CO_2$ e of carbon emissions.

(2)    Biomass Storage and Transportation

The biomass storage and transportation phase includes two processes: biomass storage and biomass transportation (Table 2).

**Table 2.** Biomass storage and transportation phase life cycle inventory data [24].

| | Sessions | Energy Consumption Species | Energy Input/MJ | Emission Content | Greenhouse Gas Emissions/kg $CO_2$ e | Cost/USD |
|---|---|---|---|---|---|---|
| Biomass storage | biomass storage | Electricity | 77 | $CO_2$ | 12.4 | 1.9 |
| Biomass transportation | Vehicle transportation | Diesel | 1120 | $CO_2$ | 93.9 | 40.7 |

During the storage of biomass, rice husk, for example, needs to be kept at a dry level (keeping the moisture content within 12%). In this case, a single cylinder dryer of 18 W with a speed of 10 r/min. to maintain the dryness of 16 t of rice hulls, 21.4 kWh of electricity input is required, bringing carbon emissions of 12.4 kg $CO_2$ e. Transporting 16 t of biomass to the biomass gasification plant requires the consumption of 33.9 L of diesel, resulting in a carbon emission of 93.9 kg $CO_2$ e (Table 3).

**Table 3.** Biomass Gasification and Power Generation phase Life Cycle Inventory Data.

| | Sessions | Energy Consumption Species | Energy Input/MJ | Emission Content | Greenhouse Gas Emissions/kg $CO_2$ e | Cost/USD |
|---|---|---|---|---|---|---|
| Biomass Gasification [25] | Biomass Gasification | Rice husk | 16 t | —— | —— | —— |
| | | Electricity | 3064 MJ | $CO_2$ | 494.5 | 76.6 |
| | | Natural gas | 20,724 MJ | $CO_2$ | 1160 | 320.8 |
| Biomass power generation [25] | Gas purification | Electricity | 3340 MJ | $CO_2$ | 539 | 83.5 |
| | Gas-fired power generation | Fuel Gas | 3333 $m^3$ | $CO_2$ | 5375 | —— |

(3)    Biomass gasification and power generation

The biomass gasification and power generation stage includes two processes: biomass gasification and biomass power generation. First, in the biomass gasification stage, 16 t of dried biomass needs to be pre-processed into similar sized pellets, requiring 851 kWh of electricity to be supplied, resulting in an indirect carbon emission of 494 kg $CO_2$ e. The generated gas is gasified to high temperature crude gas by cyclone return particles, the chemical reaction equation of this process is:

$$Biomass \rightarrow CO + H_2 + CO_2 + CH_4 + H_2O$$

Next, the biogas undergoes oxidation and reduction reactions at 600–1000 °C. The main reaction equations are

$$2CO + O_2 \rightarrow 2CO_2; C + H_2O \rightarrow CO + H_2$$

The whole process is a heat absorption reaction and therefore requires a heat source from outside, which requires the use of 583 m$^3$ of natural gas, a process that will result in 1160 kg $CO_2$ e emissions.

The biomass power generation phase consists of two processes: gas purification and gas power generation, which first requires further upgrading of the gas produced by biomass gasification through a variable pressure adsorption unit, a process that will use 928 kWh of electricity and result in indirect carbon emissions of 539 kg $CO_2$ e. In the gas-fired power generation process, 3333 m$^3$ of gas is used as the raw material for power generation, and by doing work in the gas turbine, the generator is driven to rotate and generate power, resulting in indirect carbon emissions of 5375 kg $CO_2$ e and generating 10$^4$ kWh of biomass electricity.

## 4. Results and Discussion

### 4.1. Analysis of Greenhouse Gas Emissions

The LCA assessment of biomass gasification power technology in China reveals that the whole life cycle GHG emissions of biomass gasification power technology is 8.68 t $CO_2$ e and the whole life cycle cost is 674 USD/10$^4$ kWh. Figure 2 shows the greenhouse gas emissions of biomass gasification and power generation rush full life cycle in stages, and it is found that the biomass gasification and power generation stages account for 87% of the total emissions. In the biomass gasification and power generation phase, the natural gas heat source required for the biomass pyrolysis process and gasification reaction contributes 15%, and the gas turbine power generation process contributes 71%.

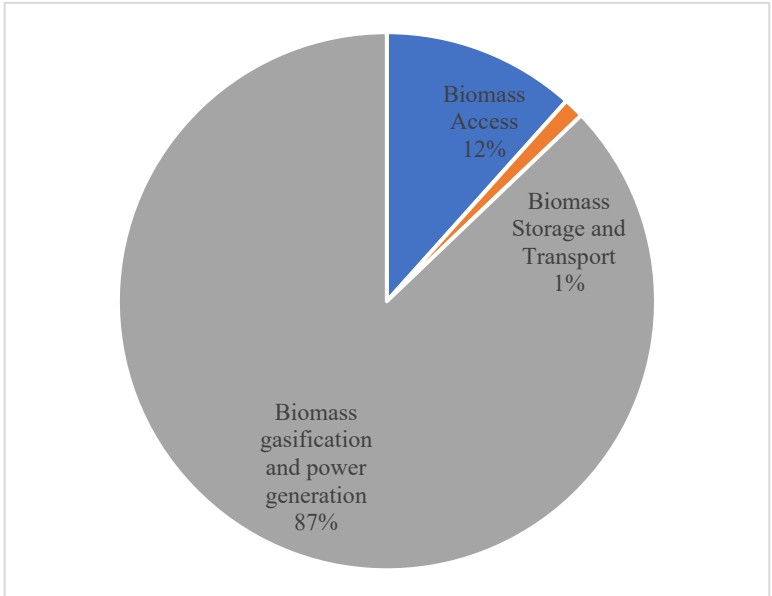

**Figure 2.** Distribution of greenhouse gas emissions by phase for biomass gasification power generation technology.

Figure 3 shows the whole life cycle carbon emissions comparison of different local power generation technologies in China. The whole life cycle carbon emission of coal power generation technology is 1.08 kg $CO_2$ e/kWh, and the whole life cycle carbon emission of photovoltaic power generation technology is 0.0984 kg $CO_2$ e/kWh [26]. The whole life cycle carbon emission of biomass gasification power technology is 0.868 kg $CO_2$ e/kWh, which is 14.7% less than that of coal power technology, but the carbon emission is still higher compared to photovoltaic power technology.

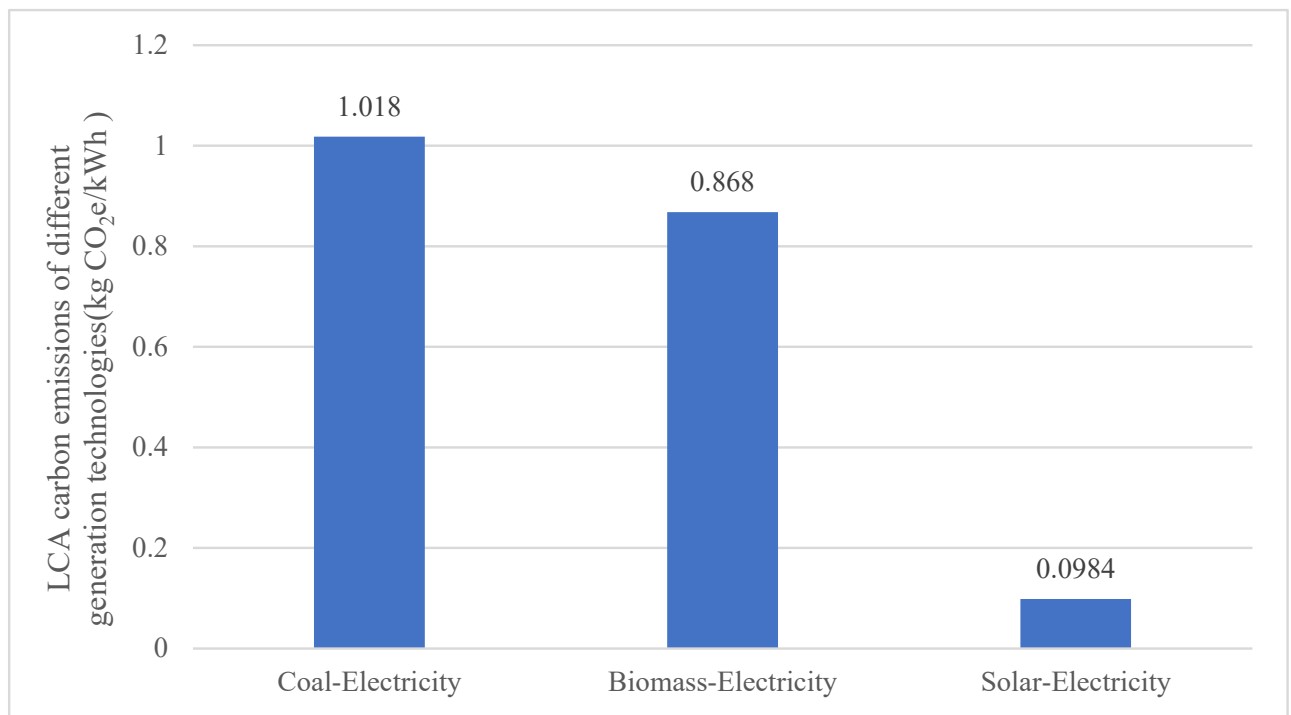

**Figure 3.** Comparison of whole life cycle carbon emissions of different power generation technologies in China (kg $CO_2$ e/kWh).

The measured GHG emissions from biomass gasification power generation in rural areas are 8.6% lower than those from biomass gasification power generation in urban areas in China. This is due to the ability to collect biomass resources in rural areas and to reduce the diesel and electricity consumption in the process with a relatively centralized transportation method. It can be seen that biomass gasification technology is more adaptable in rural areas [13].

To further reduce the whole life cycle carbon emissions of biomass gasification power technology, this paper argues that the following points need to be addressed.

First, improve the efficiency of biomass gas power generation. The current efficiency of biomass gasification power generation gas turbine is measured according to "3 $m^3$ gas to generate 1 kWh electricity", and the energy conversion efficiency of power generation is 33.3%, which is still a long way from the more mature coal power generation technology. If the efficiency of biomass gas power generation can be improved, then biomass gasification power generation does not require much biomass gas for every $10^4$ kWh of electricity generated, thus greatly reducing the indirect carbon emissions from gas combustion.

Second, reduce the carbon emission factor of the power sector. The current electricity used in the whole life cycle process of biomass gasification power generation technology in China is still from grid electricity, with a carbon emission factor of 0.581 kg $CO_2$/kWh. If more renewable energy is used in the power system, the grid emission factor can be reduced, leading to a reduction in the carbon emission factor of biomass gasification power generation technology as well.

Third, new energy vehicles are used for transportation. The current vehicles used for biomass collection and transportation are all diesel vehicles, thus leading to larger indirect carbon emissions. If diesel vehicles are replaced by electric vehicles or hydrogen heavy-duty trucks suitable for long-distance transportation, the life-cycle carbon emissions of biomass gasification and power generation technology will be further reduced.

Fourth, the Carbon Capture, Utilization and Storage (CCUS) technology is proposed to be applied in biomass gasification power generation technology. In the process of biomass gas power generation, the combustible gas combustion leads to a large amount of $CO_2$

emission. Therefore, this type of power generation also has a large environmental pollution from the perspective of carbon emission. Therefore, it is possible to use CCUS technology in the part where $CO_2$ emissions are high and store $CO_2$ as a raw material for methanol production or as a building material, which can effectively reduce the carbon emissions of LCA from biomass gasification power generation technology.

*4.2. Economic Analysis*

The cost of biomass gasification for power generation technology by phase is shown in Figure 4. It was found that the biomass gasification power generation stage resulted in 72% of the costs and the biomass acquisition stage resulted in 22% of the costs. In the biomass gasification power generation stage, the use of natural gas as an external heat source resulted in 66.7% of the cost. In the biomass acquisition stage, the high electricity consumption for urea production resulted in 50.3% of the cost.

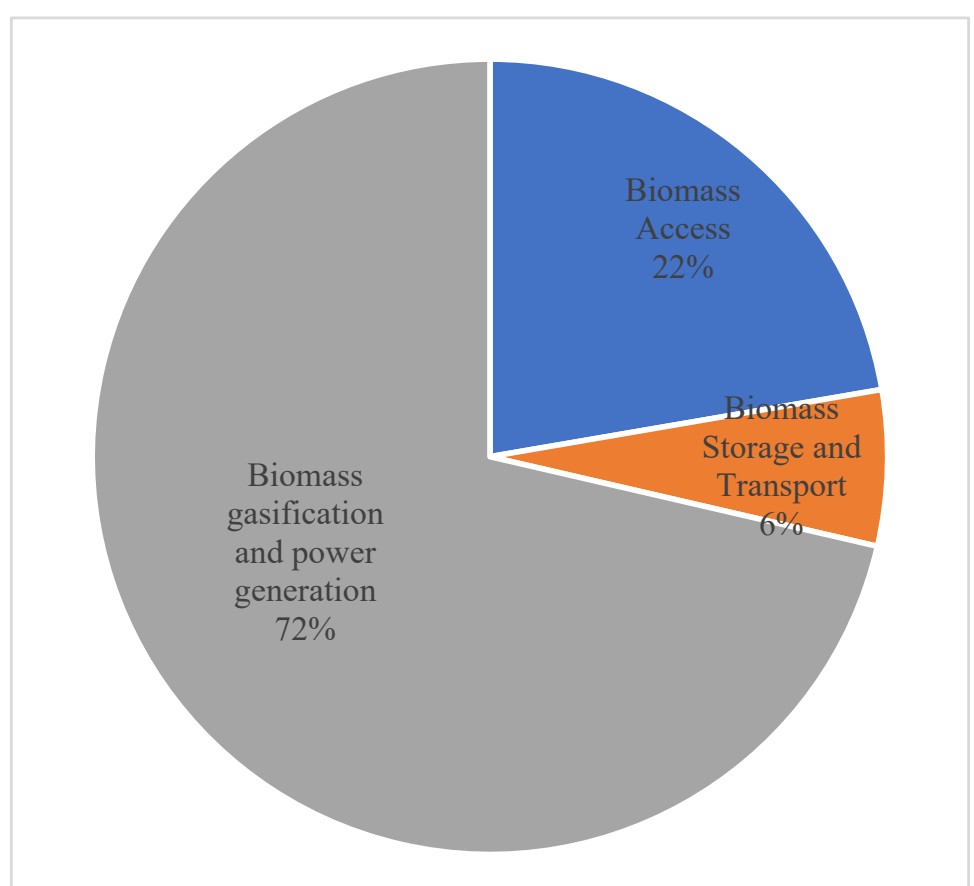

**Figure 4.** Cost distribution of biomass gasification power generation technology by phase.

The cost distribution of biomass gasification power generation technology by species is shown in Figure 5. It can be seen that the natural gas heat source that needs to be invested in the biomass gasification stage occupies a higher cost. If a lower-priced fuel is used as the heating method or the heating system of the biomass gasification process is optimized, the whole life cycle cost of biomass gasification technology can be effectively reduced. Furthermore, this paper presents the whole life cycle costing equation for biomass gasification power technology.

$$\text{Cost}_{\text{bio-e}} = 89 \times \text{Diesel Price(USD/L)} + 3564 \times \text{Natural gas prices (USD/m}^3) + 4920 \times \text{Electricity price (USD/kWh)}$$

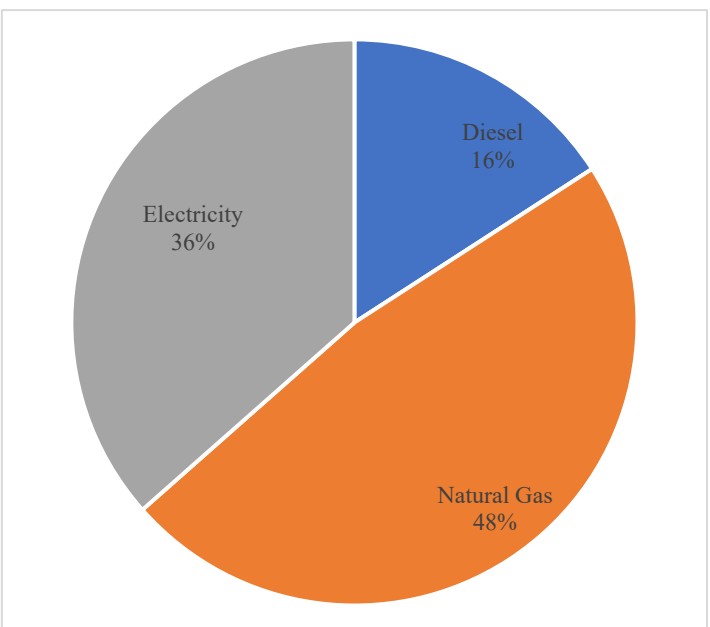

**Figure 5.** Cost distribution of biomass gasification power generation technology by species.

In terms of economics, the current cost of electricity production in China using biomass gasification technology is about 0.067 USD/kWh per unit of electricity produced, which is basically the same as the cost of coal power generation. This shows that biomass gasification power generation is very economical among all forms of power generation. It is worth mentioning that the additional cost caused by the construction of biomass gasification plant is not considered in the economical calculation of biomass gasification power generation technology, if this part of the cost is also calculated, then the current LCA cost of biomass gasification power generation technology will be higher.

In order to reduce the whole life cycle cost of biomass gasification and power generation technology, it is necessary that the mixture of gas produced in the biomass gasification and power generation phase can be used as fuel gas, and by replacing a certain percentage of natural gas heat source, the cost structure of the whole process can be reduced. In addition, from the biomass acquisition stage, reducing the energy and material consumption of urea production by improving the process of urea production is also an important way to reduce the cost of LCA from biomass gasification for power generation.

## 5. Conclusions

A large amount of waste biomass is available in China. By collecting biomass and generating biopower, biomass can be recycled and the security of China's energy system can be enhanced. In this paper, we conducted a full life-cycle assessment using LCA method with local biomass gasification power generation data in China and found that the full life-cycle GHG emissions of biomass gasification power generation technology is 8.68 t $CO_2$ e/$10^4$ kWh and the full life-cycle cost is 674 USD/$10^4$ kWh.

By analyzing the GHG emissions in stages, it was found that the "biomass gasification and power generation" stage contributes 87% of the carbon emissions, with the gas-fired power generation process contributing the most carbon emissions.

The measured GHG emissions from biomass gasification power generation in rural areas are 8.6% lower than those from biomass gasification power generation in urban areas in China. Therefore, biomass gasification power generation technology is more adaptable in rural areas. Comparing the LCA carbon emissions of coal, biomass gasification and photovoltaic technologies in China, it is found that the whole life cycle carbon emissions of biomass gasification technology are 14.7% lower compared to coal technology, but still higher compared to photovoltaic technology. In order to reduce the whole life cycle carbon

emissions of biomass gasification power generation technology, this paper proposes the following recommendations: (1) improve the efficiency of biomass gas power generation; (2) reduce the carbon emission factor in the power sector; (3) adopt new energy vehicles for transportation; (4) apply the Carbon Capture, Utilization and Storage (CCUS) technology is proposed to be applied in biomass gasification power generation technology.

In terms of economics, based on the phased cost results, it was found that the "biomass gasification and power generation" phase resulted in 72% of the costs, with natural gas as the largest additional cost for the external heat source. Therefore, in order to reduce the whole life cycle cost of biomass gasification for power generation, this paper proposes the concept of using the generated gas mixture to be able to be used as a fuel gas to replace a certain percentage of natural gas heat source. Finally, this paper proposes a whole life cycle costing formula for biomass gasification power generation technology in China through the cost decomposition of LCA, which can lay the foundation for future LCA assessment in related fields.

**Author Contributions:** Validation, Y.Y.; Investigation, Y.W.; Data curation, Y.W.; Writing—original draft, Y.W.; Writing—review & editing, Y.Y. All authors have read and agreed to the published version of the manuscript.

**Funding:** This research received no external funding.

**Institutional Review Board Statement:** Not applicable.

**Informed Consent Statement:** Informed consent was obtained from all subjects involved in the study.

**Data Availability Statement:** Not applicable.

**Conflicts of Interest:** The authors declare no conflict of interest.

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
