# Peer review of "Research on Greenhouse Gas Emissions and Economic Assessment of Biomass Gasification Power Generation Technology in China Based on LCA Method"

_sustainability, doi:10.3390/su142416729_

Round 1
Reviewer 1 Report
Kindly mention the complete forms of the acronyms while mentioning them for the first time in the abstract and the main text, e.g. life cycle assessment (LCA).
In the Abstract: the unit of the value 673.7USD seems incomplete. Please mention the basis of this value, i.e. this value is per unit capacity or per tone of CO2 emission or any other quantity.
The graphical quality of Figure 1 can be improved.
The description of the methodology should be in detail rather than contracted to a single paragraph (Section 2).
At the start of Section 3, there should be some elaboration prior to moving ahead with the subsections of different types of data.
The results and Discussion section can be made more effective and contribution by incorporating more observations and fulfilling the claims presented in the last paragraph of the Introduction section.
The same should be reflected in the clonclussion section as well.
Author Response
Thank you for your valuable comments, after researching this paper, the following changes have been made.
First, the value "673.7USD" in the summary and conclusion section has been changed to "673.7USD/10000kWh".
Second, Figure 1 of the paper has been adjusted for both fonts and graphics, and the quality has been improved.
Third, the methodological section is expanded in conjunction with the research of this paper.
Four, a relevant description is given at the beginning of the data list in Chapter 3.
At last, in the discussion section, the issues surrounding greenhouse gas emissions are discussed in the context of LCA research on biomass gasification power technology in urban areas of China.This paper finds that "the measured GHG emissions of biomass gasification power generation in rural areas are 8.6% lower than those of biomass gasification power generation in urban areas in China. This is due to the ability of rural areas to collect biomass resources in close proximity and to reduce the diesel and electricity consumption in the process with relatively centralized transportation. It is evident that biomass gasification power generation technology is more adaptable in rural areas."Based on this, the conclusion section was modified.
Reviewer 2 Report
Dear authors,
Using LCA method, this paper divided biomass gasification power generation technology into three stages: biomass acquisition, storage and transportation, gasification and power generation, calculated greenhouse gas emissions and conducted economic analysis. The topic of this paper is interesting and the authors have done a lot of work. Here are some comments:
1.There is too little content in 2.Methodology, and there is a contradiction between functional units, which should be 104kwh or 104kWh. In addition, the common unit in LCA should be 1kWh or 1MWh.
2.Where is the data of energy consumption obtained in the greenhouse gas emissions calculation process? It is necessary to explain each data source.
3.The life cycle cost in this paper only includes energy consumption, not materials and other consumption, so whether the calculation of life cycle cost is significant and comparable.
4.The language is not fluent enough. As shown below in Table 3 on page 6, there is “The biomass gasification and power generation stage includes two processes: biomass gasification and power generation” . Other places also need to be carefully checked.
Author Response
Thank you for your valuable comments, after researching this paper, the following changes have been made.
First, the methodological section is expanded in conjunction with the research of this paper.
“As shown in Figure 1, in order to obtain 10,000kWh of biomass electricity, three stages are required: biomass access, biomass storage and transport, and biomass gasification and power generation. In the whole life cycle process, the material and energy inputs are reflected in the "input" , and the greenhouse gas emissions such as CO2 are reflected in the "output".”The functional unit for LCA calculations in the paper has been standardized to "104kWh".
Second, The energy consumption data and emission data sources in the LCA data inventory have been described in the paper.
Third, this paper performs a full life-cycle operating cost accounting, which only includes the cost of energy consumption involved in the process and does not include the cost due to material inputs such as infrastructure. This is due to the fact that, for producing 104 kWh of biomass electricity, the material inputs have minimal impact on the full life cycle. Biomass resources, such as straw, are inherently recycled and reused materials and are therefore not included in the costing.
At last,The sentences in Table 3 have been modified to “The biomass gasification and power generation stage includes two processes: biomass gasification and biomass power generation. ”Other parts of the article are also grammatically polished.
Reviewer 3 Report
Remarks related to the writing of the paper
Non-unitary writing of the work
It is used like this, in many places in the work:
- Both kwh and kWh (the correct form for the electrical energy measurement unit); - m3 as well as m3 (the correct form for the volume measurement unit);
- CO2e as well as CO2 e (provided that the first form of writing would be correct);
- On page 2 it is written "Song evaluated biomass gasification power generation technology for municipal waste in Macao and found that its GHG emissions were about 0.95 kg kg CO2/kwh", where the unit kg appears twice;
- "The functional unit of the biomass gasification power generation process studied in this paper is 104kwh of electricity" is stated on page 3 before Figure 1. Then, in Figure 1 it is written 104 kwh (which is completely different);
- Each table presented in chapter 3 is marked with Table 3 (the logic would be the notation Table 1, Table 2, Table 3 or even Table 3.1, Table 3.2, Table 3.3);
- The notation for fuel consumption (diesel) is also used, writing with a capital letter "L", for liter (correctly a notation with a small letter), while for other units of measure a correct writing is used (kg, for example, for the table);
- The writing of numbers with decimals is unevenly used (for example: 3,333 m3 and 673.7 USD). Moreover, it is written in the form of 10,000kWh, an electrical energy that was stated to be 104 kWh, i.e. 10000 kWh.
Important remarks related to the collection of information from the tables in chapter 3
- In the first table, a bibliographic source is indicated from where the information was collected, only for Agricultural farming respectively (DONG Long-li, 2014), but not for Biomass collection;
- The second table related to Biomass storage and transportation indicates as the source of information: Ministry of Environmental Protection, 2019;
- The third table no longer specifies any bibliographic source (the one referring to Biomass Gasification and Power Generation).
This information is vital in the economy of the work and the sources from where these data were collected must be presented.
Author Response
Thank you for your valuable comments, after researching this paper, the following changes have been made.
First, The units in this article have been revised. For example, the unit of electricity Was corrected to "kWh" and the unit of "L" Was changed to "l".
Second, the sentences in the literature review section have been revised.第
Third, the table name in Chapter 3 has been changed to "Table1." "Table2." "Table3."
At last, The functional unit for LCA calculations in the paper has been standardized to "104kWh".
The energy consumption data and emission data sources in the LCA data inventory have been described in the paper. The article were also grammatically polished.
Round 2
Reviewer 1 Report
The authors have incorporated the given suggestions. The manuscript is recommended for Acceptance.
Reviewer 2 Report
The revision is satisfactory.